# Assessment of Serum Neopterin as a Biomarker in Peripheral Artery Disease

**DOI:** 10.3390/diagnostics11101911

**Published:** 2021-10-15

**Authors:** Agnieszka Zembron-Lacny, Wioletta Dziubek, Anna Tylutka, Eryk Wacka, Barbara Morawin, Katarzyna Bulinska, Malgorzata Stefanska, Marek Wozniewski, Andrzej Szuba

**Affiliations:** 1Department of Applied and Clinical Physiology, Collegium Medicum University of Zielona Gora, 65-417 Zielona Gora, Poland; a.zembron-lacny@cm.uz.zgora.pl (A.Z.-L.); a.tylutka@cm.uz.zgora.pl (A.T.); b.morawin@cm.uz.zgora.pl (B.M.); 2Faculty of Physiotherapy, University School of Physical Education in Wroclaw, 51-612 Wroclaw, Poland; wioletta.dziubek@awf.wroc.pl (W.D.); katarzyna.bulinska@awf.wroc.pl (K.B.); malgorzata.stefanska@awf.wroc.pl (M.S.); marek.wozniewski@awf.wroc.pl (M.W.); 3Collegium Medicum, University of Zielona Gora, 65-417 Zielona Gora, Poland; 4Department of Angiology, Hypertension and Diabetology Medical University Wroclaw, 50-556 Wroclaw, Poland; andrzej.szuba@umed.wroc.pl; 5WROVASC An Integrated Cardiovascular Centre, Specialist District Hospital in Wroclaw, Centre for Research and Development Wroclaw, 51-124 Wroclaw, Poland

**Keywords:** inflammation, nitro-oxidative stress, endothelial progenitor cells, hematopoietic progenitor cells, physical performance

## Abstract

Neopterin (NPT), a pyrazino-pyrimidine compound mainly produced by activated macrophages, has been regarded as a proinflammatory and proatherosclerotic agent. The study was designed to evaluate NPT level and its interaction with conventional peripheral artery disease (PAD) biomarkers and vascular regenerative potential in severe PAD. The study included 59 patients (females *n* = 17, males *n* = 42) aged 67.0 ± 8.2 years classified into two groups based on ankle-brachial index (ABI) measurements (ABI ≤ 0.9 *n* = 43, ABI ≤ 0.5 *n* = 16). A total of 60 subjects aged 70.4 ± 5.5 years (females *n* = 42, males *n* = 18) with ABI > 0.9 constituted a reference group. NPT concentration reached values above 10 nmol/L in patients with PAD, which differed significantly from reference group (8.15 ± 1.33 nmol/L). High levels of CRP > 5 mg/L, TC > 200 mg/dL as well as lipoproteins LDL > 100 mg/dL and non-HDL > 130 mg/dL were found in the same group, indicating the relationship between NPT and conventional atherogenic markers. The endothelial progenitor cells (EPCs) tended toward lower values in patients with ABI ≤ 0.5 when compared to reference group, and inversely correlated with NPT. These findings indicate a crucial role of NPT in atheromatous process and its usefulness in monitoring PAD severity. However, the role of NPT in chronic PAD needs further studies including relatively high number of subjects.

## 1. Introduction

Endothelial dysfunction is an essential factor preceding the development of peripheral artery disease (PAD) including chronic lower limb ischemia. PAD represents a major public health problem due to its high and increasing prevalence, worldwide distribution, and significant morbidity and mortality rates. The prevalence of PAD increases with age, especially in individuals over 65 years old. Approximately 10–25% of people over 55 years of age and 40% of people aged over 70 years suffer from this disease. With a growing aging population, the prevalence of PAD is likely to increase even further [1]. Female gender is a risk factor for PAD with apparently higher rates in low- and middle-income countries, whereas in high-income countries, the male gender tends to be an independent risk factor for PAD [2].

There have been many attempts to investigate endothelial function including nitro-oxidative stress and lipoprotein oxidation, apoptotic and inflammatory mediators, etc. [1,3,4]. Endothelial inflammation stimulates the production of pro-inflammatory cytokines and adhesion molecules such as interleukin 6, adhesion molecules (MCP-1, ICAM-1, VCAM-1), and E-selectin. These effectors support monocyte and T-cell adhesion and infiltration into the neointima lesion, and then the secretion of pro-inflammatory cytokines such as tumour necrotic factor α (TNFα) and interferon γ (INFγ) from these cells. Monocyte-derived macrophages phagocytate oxidized low-density lipoproteins (oxLDL) to become foam cells, which depends on the balance between the uptake of oxLDL and the efflux of free cholesterol [5,6,7]. Apart from the accumulation of macrophage foam cells, the migration and proliferation of vascular smooth muscle and endothelial progenitor cells (EPCs), and the production of extracellular matrix components such as the collagens, fibronectin, and elastin, metalloproteinases and their tissue inhibitors, all contribute to the atherosclerotic plaque development [5,6,8]. Recently, some attention has been focused again on neopterin (NPT) produced by activated macrophages and dendritic cells upon stimulation with IFNγ released by T cells. Neopterin is a pyrazino-pyrimidine compound that may play an important role in the pathogenesis and progression of coronary artery diseases (CAD) and peripheral artery diseases by regulating nitric oxide (NO) bioavailability and the proliferation and differentiation of hematopoietic stem cells [6,9,10]. The first studies on NPT in relation to atherogenesis and myocardial infarction were already published in the 1990s [11,12]. Lanser et al. [13] established that serum NPT level is associated with cardiomyopathy severity and could be an independent predictor of prognosis in patients with heart failure. According to Avanzas et al. [14], elevated serum NTP predicts future adverse cardiac events in patients with chronic stable angina pectoris. Furthermore, some studies demonstrated an association of NPT with the ankle-brachial index (ABI) and increased mortality of patients with vascular calcification or patients undergoing angiography [10,15,16]. Contrastingly, Shirai et al. [6] reported that NPT level increased in coronary lesions and circulating blood in patients with CAD to counter inflammation and atherosclerosis. On the basis of the gathered data on the effects of NPT on atherogenesis and its expected diagnostic usefulness, this study was designed to evaluate circulating NPT and its interaction with conventional endothelial dysfunction markers and vascular regenerative potential in a patient population with peripheral artery diseases who were stratified by ABI.

## 2. Material and Methods

### 2.1. Subjects

The study was conducted in 59 patients (females *n* = 17, males *n* = 42) aged 67.0 ± 8.2 years with chronic femoral popliteal ischemia (*n* = 45), aorto-iliac ischemia (*n* = 9), multilevel (*n* = 3), and peripheral ischemia (*n* = 2), at a routine follow-up visit at WROVASC Research Centre (Table 1). A detailed cardiovascular examination was performed and the demographic data on cardiovascular risk factors such as hypertension, diabetes, hyperlipidaemia, family history of cardiovascular diseases, and present smoking habits were recorded for every patient during the visit in WROVASC Research Centre. Medications taken by the PAD patients are presented in Table 2. The following inclusion criteria were applied: age > 60 years, documented, stable, at least 3-month intermittent claudication, and ABI ≤ 0.9. The exclusion criteria included acute infectious diseases, uncontrolled hypertension and/or diabetes, oncologic diseases, neurodegeneration diseases, or revascularization procedure in the past 3 months based on the assessment of an experienced physician. All the recruited patients were diagnosed with chronic leg ischemia at Fontaine stage II (A and B) [17]. The majority of our patients had femoropopliteal occlusion, however, some had mostly peripheral below the knee lesions. A total of 60 patients, females (*n* = 42) and males (*n* = 18), with no history of PAD and ABI >0.9 constituted a reference group (controls; Table 1). The medications taken by the control group included antihypertensive (84%) and hypolipidemic (10%) drugs as well as anticoagulants including anti-platelet agents (15%). Approximately 15% of the study subjects suffered from a stable coronary artery disease and took medications in prevention of thromboembolic events. Cigarette smokers were not included in the study. Diabetes accounted for 34% (*n* = 20) of PAD patients and 12% (*n* = 7) of the control group. All the subjects were informed of the aim of the study and signed a written consent to participate in the project. The protocol of the study was approved by the Bioethics Commissions at Medical University of Wroclaw, Poland (N° KB-130/2008) and at Regional Medical Chamber Zielona Gora, Poland (N° 01/66/2017), in accordance with the Helsinki Declaration.

### 2.2. Hemodynamic Analysis

The ankle-brachial index was measured with a sphygmometer and a hand-held pocket Doppler device (Doppler MD2, 8 MHz, Huntleigh Healthcare, Cardiff, UK) by an experienced physician. Following a 15-min rest period in a supine position, the brachial and ankle systolic pressure levels were determined using appropriately sized cuffs placed on the brachial artery and on the ankle above the malleoli. To calculate the ABI of each leg, the higher value of the ankle pressure measured on dorsalis pedis and tibialis posterior for each limb was used in the numerator, and the higher brachial pressure level measured on both arms was used in the denominator. The patients were categorized into two groups based on their ABI measurements; the patients with low ABI ≤ 0.9 (mild-moderate disease severity) and patients with very low ABI ≤ 0.5 (high level of the disease severity) according to Arain et al. [18].

### 2.3. Physical Performance

The 6-min walk test (6MWT) was performed according to the technical standards of American Thoracic Society [19]. The total distance walked in the test was measured and the 6MWT gait speed was then calculated by the following equation: 6 MWT gait speed (m/s) = total distance (m)/360 s [2]. Following the classification by Middelton et al. [20], a gait speed within the range of 1.0 to 1.3 m/s classified the older adults as active while a gait speed < 1.0 m/s classified them as inactive.

### 2.4. Blood Sampling

Fasting blood samples were collected from the median cubital vein in the morning between 8.00 and 10.00 using S-Monovette tubes (Sarstedt, Austria). The whole blood samples were placed into specimen tubes containing EDTA and were immediately analysed in professional laboratory company Diagnostyka (Poland, ISO15189). For the other biochemical analyses, blood samples were centrifuged at 3000 rpm for 10 min, and aliquots of serum were stored at −80 °C. The average intra-assay coefficients of variation (intra-assay CV) for the used ELISA kits were <10%. All samples were analysed in duplicate or triplicate in a single assay to avoid inter-assay variability.

### 2.5. Atherogenic Markers

Triglycerides (TG), total cholesterol (TC), high-density lipoproteins (HDL) and low-density lipoproteins (LDL) were determined by laboratory company Diagnostyka (Poland, ISO15189). The non-HDL cholesterol was calculated by subtracting HDL from the total cholesterol concentration. Oxidized low-density lipoprotein (oxLDL) and 3-nitrotyrosine (3NT) were determined by using ELISA kits from SunRed Biotechnology Company (Shanghai, China) with detection limit 30.3 ng/mL and 0.007 nmol/mL, respectively.

### 2.6. Inflammatory Markers and Progenitor Cells

Neopterin (NPT) and C-reactive protein (CRP) concentrations were determined using commercial ELISA kits from DRG International Inc. (USA). The detection limits for NPT and CRP were set at 0.7 nmol/L and 0.001 mg/L, respectively. CD38 level, as a marker of non-hematopoietic cells, was determined by using ELISA kits from SunRed Biotechnology Company (China) with detection limit of 0.009 ng/mL. Endothelial progenitor cells (EPCs) and CD34, as a marker of hematopoietic progenitor cells, were determined by using ELISA kits from SunRed Biotechnology Company (Shanghai, China). The detection limits for EPCs and CD34 were 0.125 ng/mL and 0.167 ng/mL, respectively.

### 2.7. Statistical Analysis

Statistical analyses were performed using the R version 4.0.3 [21]. The assumptions for the use of parametric or non-parametric tests were checked using the Shapiro-Wilk and the Levene tests to evaluate the normality of the distributions and the homogeneity of variances, respectively. The significant differences in mean values between the groups were assessed by the one-way ANOVA and the Tukey’s post hoc test. If the normality and homogeneity assumptions were violated, the Kruskal-Wallis non-parametric test (ABI ≤ 0.9, ABI ≤ 0.5 and control group) or Mann-Whitney non-parametric test (females vs. males) was used. Additionally, eta-squared (η^2^) based on function in RStudio was used to measure of effect size which is indicated as having no effect if 0 ≤ η^2^ < 0.01, a small effect if 0.01 ≤ η^2^ < 0.06, a moderate effect if 0.06 ≤ η^2^ < 0.14, and a large effect if η^2^ ≥ 0.14. Spearman’s rank correlation (*r_s_* Spearman’s rank correlation coefficient) was used to investigate the relationship between atherogenic and inflammatory markers and PAD severity. Statistical significance was set at *p* < 0.05.

## 3. Results

### 3.1. Subjects

The study participants suffered from a moderately severe PAD with a mean ABI of 0.61 ± 0.14. Approx. 27% of patients demonstrated a very low ABI 0.44 ± 0.07. All patients reported that they experienced intermittent claudication in daily life. The maximum claudication distance (MCD) was 107 ± 55 m and did not relate to ABI or any biochemical PAD risk factors.

### 3.2. Physical Performance

On average, the patients covered the distance of 356 ± 54 m in 6MWT at a gait speed of 0.99 ± 0.18 m/s. In the 6MWT the patients with ABI ≤ 0.9 achieved a result of 376 ± 57 m (1.05 ± 0.14 m/s) while a distance of 304 ± 56 m (0.85 ± 0.16 m/s) was achieved by patients with ABI ≤ 0.5. Approx. 98% of the PAD patients achieved the result of gait speed below the reference value of 1.3 m/s, which proves the problems with walking due to leg ischemia resulting in lower physical activity. The positive correlations between ABI and 6MWT (*r_s_* = 0.410, *p* < 0.05) and between ABI and the gait speed (*r_s_* = 0.422, *p* < 0.05) were observed. The control group demonstrated an active lifestyle and approx. 60% of them achieved the result of gait speed above 1.3 m/s.

### 3.3. Atherogenic Markers

High levels of TC > 150 mg/dL, LDL > 130 mg/dL and non-HDL > 130 mg/dL were found in 25% of patients and in 50% of control group with 70% of patients and only 10% of controls taking the hypolipidemic drugs. Among patients with ABI ≤ 0.9 and ABI ≤ 0.5, no difference was recorded in their lipoprotein-lipid profile. The control group demonstrated higher values compared to patients specially for LDL and HDL. Contrastingly, oxLDL and 3NT concentrations were significantly lower in controls possibly due to their daily physical activity which improves nitro-oxidative metabolism. Both markers of oxidative stress changed in parallel (oxLDL/3NT *r_s_* = 0.561, *p* < 0.05). The value η^2^ indicates a large effect of PAD on cholesterol metabolism and 3NT that is a marker of NO bioavailability (Table 3). Interestingly, higher concentrations of oxLDL were detected in female patients (females 769 ± 545 ng/mL and males 489 ± 294 ng/mL) and female controls (females 404 ± 290 ng/mL and males 119 ± 130 ng/mL) than in males in both groups (Figure 1). Although there were significant differences identified in the individuals without PAD, the results suggest that female gender increases the risk of LDL oxidation and thus atherogenesis in old age. OxLDL and 3NT stimulated an increase in CD38 that is also described as a prognostic marker in endothelial dysfunction [22]. These relationships were demonstrated by correlations oxLDL/CD38 *r_s_* = 0.519 (*p* < 0.001) and 3NT/CD38 *r_s_* = 0.401 (*p* < 0.01). There is a large body of evidence that oxLDL and 3NT could act on endothelial cells activities in almost every aspect, such as proliferation, differentiation, apoptosis, mobilization, migration, and senescence [23].

### 3.4. Inflammatory Markers and Progenitor Cells

High levels of CRP > 5 mg/L were found in 30% of patients with ABI ≤ 0.9 and in 100% of patients with ABI ≤ 0.5. NPT concentration reached the values > 10 nmol/L in the same group of patients, indicating the association of NPT with CRP as a circulating biomarker in PAD severity. NPT concentration highly correlated with CRP and ABI (Figure 2 and Figure 3) and conventional atherogenic markers but demonstrated an inverse correlation with ABI and endothelial progenitor cells (Table 4). This indicates a crucial role of NPT in atheromatous process but simultaneously excludes NPT from its participation in vascular regeneration. The levels of EPCs, CD34, and CD38 were significantly lower in ABI ≤ 0.5 group which proves their impaired vascular regenerative potential in severe PAD. Interestingly, CD34 and CD38 were significantly higher in the study females (25.78 ± 13.84 ng/mL and 1.186 ± 0.709 ng/mL) than males (18.57 ± 9.51 ng/mL and 0.790 ± 0.392 ng/mL) (Figure 4 and Figure 5). The same tendency was observed in the control group for CD34 and CD38 (females 22.43 ± 11.03 ng/mL and 1.283 ± 0.884 ng/mL; males 18.36 ± 7.82 ng/mL and 0.722 ± 0.120 ng/mL) (Figure 4 and Figure 5). A marked effect of PAD severity on NPT, CRP and EPCs levels was also proven by the high value of η^2^ (Table 5). Whether low levels of EPCs are useful in screening for PAD is an issue which needs to be investigated.

## 4. Discussion

Peripheral artery disease is a manifestation of atherosclerosis with a poor prognosis. It is generally complicated by nitro-oxidative stress and the inflammation plays an important role in the development and progression of PAD. A number of studies have investigated the association of various oxi-inflammatory mediators with atherosclerosis. NPT has already been considered as a new biomarker of macrophage activity in atherosclerosis and a strong predictor of atherosclerotic plaque instability [10,24]. The present study has focused on the strength of the relationship between NPT and other variables that determine the severity of PAD and endothelial dysfunction (Table 4). The obtained results showed the highest NPT concentration in patients with ABI ≤ 0.5 and a significant effect size of PAD severity on NPT (η^2^ > 0.14). NPT reached values above 10 nmol/L in patients with ABI ≤ 0.5. The same group of patients also demonstrated high levels of CRP > 5 mg/L, TC > 200 mg/dL as well as LDL > 100 mg/dL and non-HDL > 130 mg/dL. Especially strong association was observed for NPT with CRP, which is one of the most widely studied inflammatory molecules for PAD evaluation. Several actions are induced by CRP, and they include the release of chemoattractants, which, in turn, attracts monocytes towards the endothelial barrier. CRP also upregulates the release of the pro-inflammatory cytokines and inhibits the release of nitric oxide which precedes the development of arteriosclerosis. The association between CRP and the development of PAD was proved in a large, cross-sectional family-based study [25]. In our study, the serum levels of NPT and CRP were found to increase progressively as the ABI decreased in PAD patients. These results provide the grounds to postulate that NPT can be a novel biomarker and/or biological effector in PAD pathophysiology. Whether this candidate will improve the risk estimation associated with traditional PAD risk factors is still unclear and will need further evaluation, especially by inclusion of the biomarkers and clinical parameters described by Saenz-Pipaon et al. [1].

The high level of NPT was also associated with a low physical performance in patients with PAD, which impairs the quality of life and also increases the risk of progression towards critical limb ischemia. Chen et al. [26] were the first to propose the use of the 6MWT (and gait speed) in patients with PAD to estimate walking endurance and the improvement in ambulation among patients with PAD who participated in exercise-based intervention programs. In our study, the gait speed was 1.05 ± 0.14 m/s in patients with ABI ≤ 0.9 and 0.85 ± 0.16 m/s in patients with ABI ≤ 0.5. Approx. 98% of the PAD patients achieved the result of gait speed below the reference value of 1.3 m/s which is associated with great PAD severity.

The high concentration of NPT reveals the involvement of the activated macrophages in vascular rebuilt, which, in turn, is able to promote nitro-oxidative stress [10]. The concentrations of nitro-oxidative stress markers, such as oxLDL and 3NT, were approx. 50–100% higher in patients with PAD compared to controls. OxLDL has been shown to play a key role in atherosclerosis. In the early stages of atherogenesis, oxLDL promotes endothelial injury, induces expression of adhesion molecules, and attracts macrophages [27]. In the subintimal space of the arterial vessel wall, scavenger receptor uptake of oxLDL by macrophages is viewed as a critical step in foam cell formation. OxLDL may also contribute to the destabilization of the growing plaque and to the formation of thrombi by inducing matrix metalloproteinases. Serum levels of oxLDL have already been shown to be elevated in patients with established CAD [28]. However, compared to analyses of CAD, there are fewer studies examining the role of oxLDL in the pathogenesis of PAD [29]. In the Edinburgh Artery Study, circulating lipid peroxides were found to be higher in PAD compared to the control group, and every 1 μmol/L increase in lipid peroxides was associated with a 17%-increase in the risk of PAD [30]. Among 62 individuals with early-onset PAD requiring surgical intervention before the age of 50, circulating antibodies against oxLDL were reported to differentiate patients from controls better than conventional atherogenic markers [31]. The Bruneck prospective study followed 1510 men and women aged 40–79 years with ultrasound measures of carotid and femoral atherosclerosis and measured the concentration of antibodies against oxidized lipids in more than 90% of participants [32]. OxLDL also leads to the generation of reactive oxygen species (ROS) including peroxynitrite and superoxide whose excessive concentrations lead to a state of nitro-oxidative stress and promote atherogenesis. The ROS overproduction as well as nitration of many proteins decrease enzyme activity, disrupt metabolism and cellular detoxification, perturb cytoskeletal organization, and ultimately contribute to the cytotoxic effects of peroxynitrite. 3NT measurement is required to confirm the presence of peroxynitrite in blood [33]. In our previous paper we argued that the detection and quantification of 3NT could be used as an indicator for the pathological processes in vascular endothelium [34].

Endothelial dysfunction is a key process leading to atherosclerosis and PAD, and the organism tries to counterbalance its progress by activating EPCs mobilization and homing to the sites of vessel injury to induce repair. EPCs mobilization from the bone marrow is mainly triggered by inflammation [1]. As such, Morishita et al. [35] investigated the pattern of EPCs mobilization and their association with inflammation and nitro-oxidative stress markers in patients with PAD. We observed an increase in the number of circulating EPCs in the moderate phases of PAD which then decreased in the advanced phases of the disease, and was negatively correlated with NPT and CRP concentrations. A significant impact of PAD severity on NPT, CRP and EPCs levels was proven by the value η^2^. Progenitor cells CD34 behaved similarly to EPCs; i.e., they decreased in patients with ABI ≤ 0.5. A reduced number of circulating progenitor cells and their impaired activity, measured by colony forming and migration assays, have been reported in CAD in some yet not consistently in all studies [36,37,38]. However, compared to the analyses of CAD, there are fewer studies examining the role of progenitor cells in the pathogenesis of PAD [39]. CD34 is a transmembrane protein that was first identified on hematopoietic stem and progenitor cells. CD34 is involved in cell-to-cell adhesion as a ligand for E-selectin in regulating the tightness of endothelial cell adhesion to adjoining cells. For therapeutic reendothelialization, CD34 antibody-coated stents have been used clinically to capture circulating EPC at the injury sites and to enhance reendothelialization and safety of stents [40]. Hayek et al. [40] observed that PAD was associated with low CD34 counts. We identified low CD34 level in the advanced phases of artery disease compared to the study subjects with moderate PAD and no PAD at all who demonstrated CD34 level higher by 30%. These findings indicate an important role of circulating progenitor cells in the pathogenesis of atherosclerosis. CD38 is also a transmembrane protein that was originally described as a cell-surface marker for classification and phenotypic identification of white blood cells. CD38 can function either as a receptor or as an enzyme (known as cyclic ADP ribose hydrolase) in cell adhesion, signal transduction, and calcium signalling. Recently, its expression has been characterized in the heart and endothelial cells. CD38 activation can cause endothelial dysfunction and it increases susceptibility to atherosclerosis [20]. Boslett et al. [41] demonstrated that CD38 was activated by hypoxia-reoxygenation in endothelial cells, which causes nicotinamide adenine dinucleotide phosphate (NADP(H)) consumption along with the loss of NO generation. In our study, CD38 was found to be significantly related to low NO bioavailability, demonstrated by high concentration of 3NT in patients with PAD. Interestingly, a change tendency of CD38, similar to CD34, was observed and it was significantly higher in females than males (Figure 3). A significant gender difference in the incidence of cardiovascular diseases in humans have already been reported [42]. The role of 17β-oestradiol in the cardiovascular system is very complex. Although it has been shown that 17β-oestradiol can increase CD38 expression [43,44], there is no report available to clarify whether 17β-oestradiol can increase the CD38 expression in the endothelial cells. Moreover, there is no evidence to prove that the physiological levels of 17β-oestradiol are sufficient to increase CD38 expression to a higher level in women than in men, and whether the expression of CD38 differs between genders has not yet been proven either. It is difficult to confidently predict the changes in CD38 expression in postmenopausal women. While decreased 17β-oestradiol levels result in decreased expression and activity of CD38, the process of ageing is associated with an increase in the expression and activity of CD38. There is no research which could determine which item dominates CD38 expression in postmenopausal women [22].

## 5. Conclusions

PAD is a multifactorial disease and a single biomarker determination might not be able to fully reflect the complex pathophysiological processes underlying vascular remodelling [45]. Moreover, different inflammatory proteins and oxidants might represent distinct molecular pathways operating through different mechanisms. It has been proposed that a multi-marker approach might be more useful for PAD evaluation [1]. NPT exhibits high serum levels in chronic peripheral atherosclerotic diseases and impairs the vascular regenerative potential. Therefore, the analysis of circulating EPCs might offer a new therapeutic target in PAD, while NPT synergistically with CRP and other conventional PAD biomarkers might provide useful information on the atherosclerosis initiation and progression.

## 6. Limitation

The limitations of the study include a relatively small number of patients with PAD, especially group with ABI ≤ 0.5, and the prognostic value of inflammatory markers and progenitor cells was limited. Therefore, further randomized studies based on large populations are needed to provide stronger evidence for predicting patient outcomes.

## Figures and Tables

**Figure 1 diagnostics-11-01911-f001:**
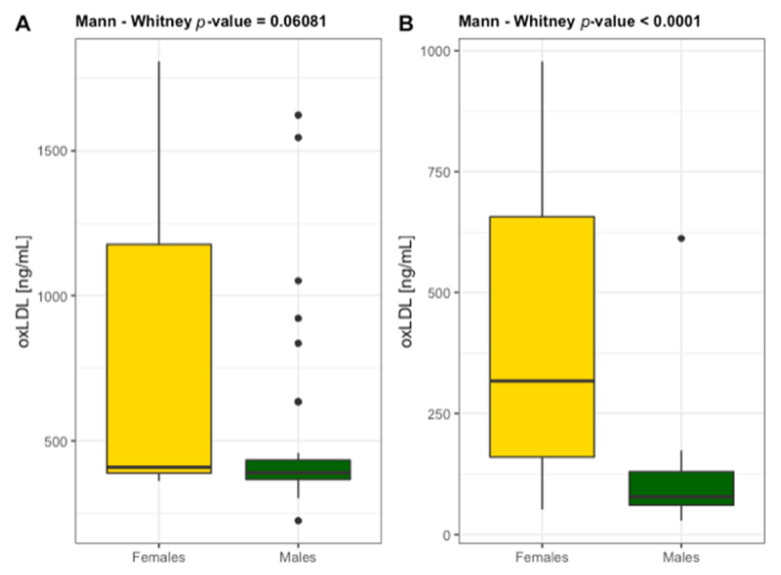
Differentiation of the concentrations of oxidized low-density lipoprotein (oxLDL) in females and males diagnosed with (**A**) *n* = 59 and without PAD (**B**) *n* = 60.

**Figure 2 diagnostics-11-01911-f002:**
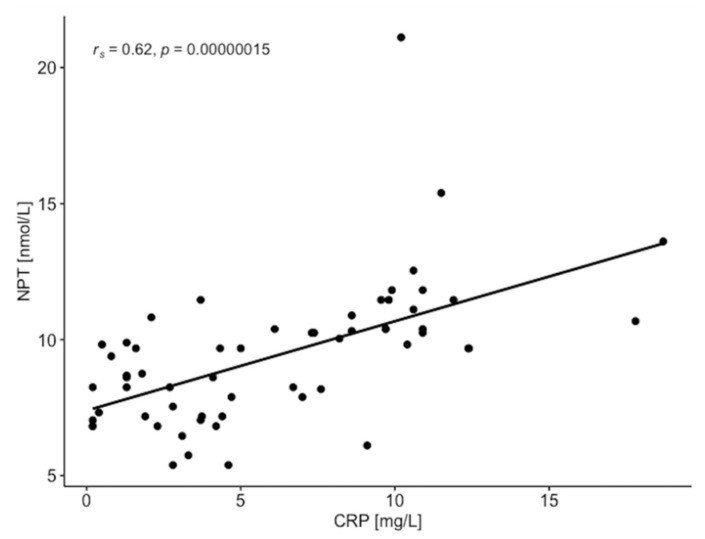
Relationship between C-reactive protein (CRP) and neopterin (NPT) in patient with peripheral artery disease (*n* = 59).

**Figure 3 diagnostics-11-01911-f003:**
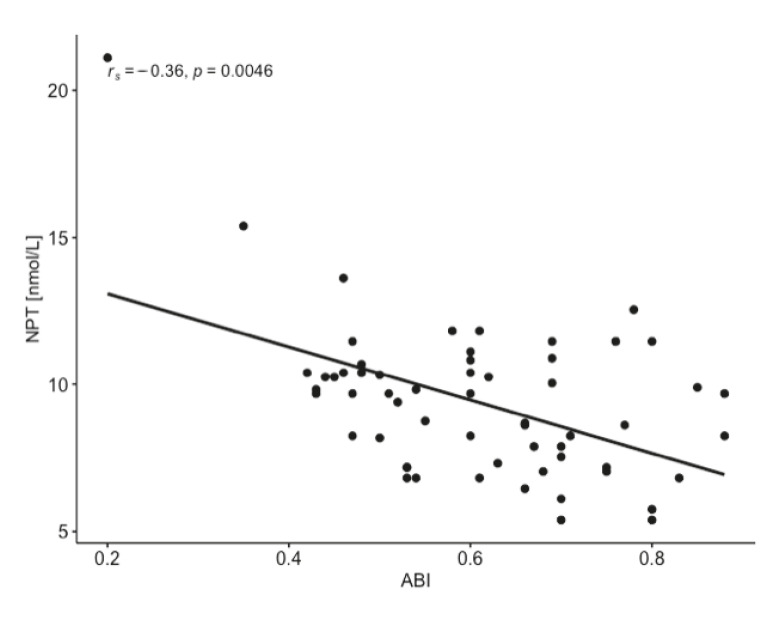
Relationship between ankle-brachial index (ABI) and neopterin (NPT) in patients with peripheral artery disease (*n* = 59).

**Figure 4 diagnostics-11-01911-f004:**
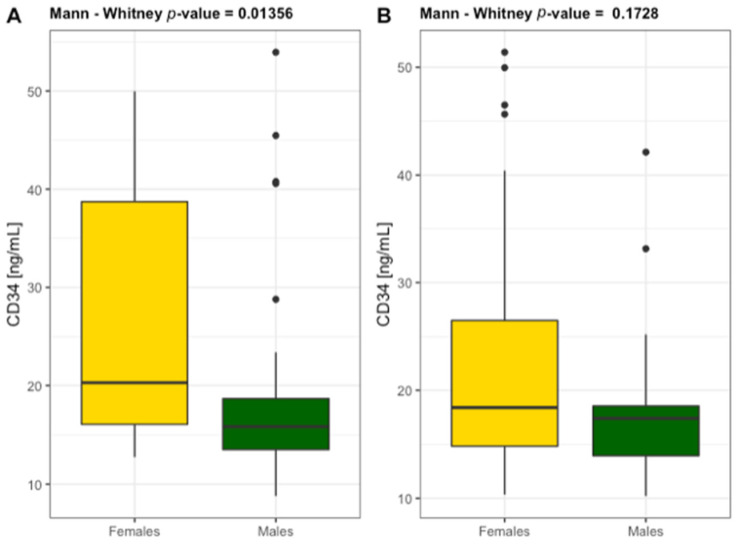
Differentiation of the level of hematopoietic progenitor cells CD34 in females and males diagnosed with PAD (**A**) *n* = 59 and without PAD (**B**) *n* = 60.

**Figure 5 diagnostics-11-01911-f005:**
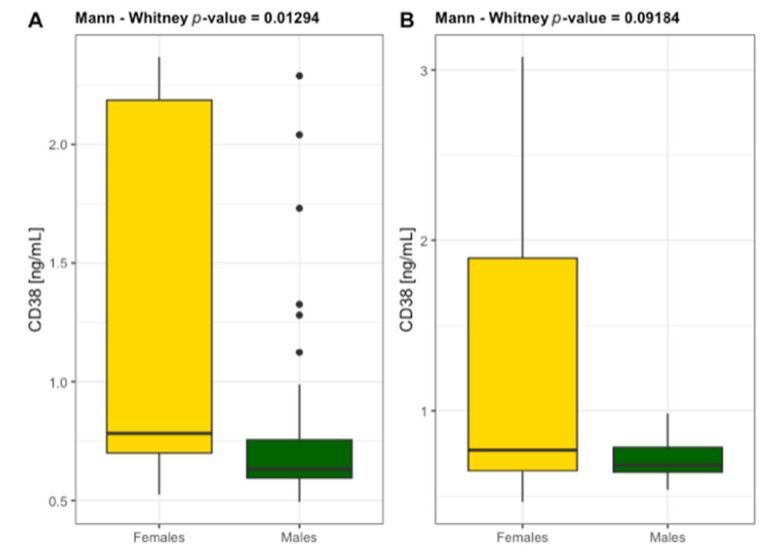
Differentiation of the level of non-hematopoietic cells CD38 in females and males diagnosed with PAD (**A**) *n* = 59 and without PAD (**B**) *n* = 60.

**Table 1 diagnostics-11-01911-t001:** Characteristics of the patients (mean ± SD).

	Patients			Controls			Patients vs. Controls	
Characteristics	Females *n* = 17	Males *n* = 42	Females vs. Males	Females *n* = 42	Males *n* = 18	Females vs. Males	Females	Males
Age [y]	68.5 ± 7.4	66.4 ± 8.5	0.387	68.9 ± 4.7	73.9 ± 5.5	<0.01	0.349	<0.01
Height [cm]	158.7 ± 5.4	171.6 ± 7.1	<0.001	160.6 ± 4.7	168.5 ± 4.7	<0.0001	0.197	0.077
Weight [kg]	66.4 ± 11.0	82.3 ± 13.0	<0.001	66.1 ± 6.7	80.0 ± 8.4	<0.001	0.902	0.051
BMI [kg/m^2^]	26.3 ± 4.0	27.9 ± 3.5	0.159	25.8 ± 2.7	27.9 ± 2.6	<0.5	0.644	0.921
Diabetes, *n* (%)	7 (41.2)	13 (30.9)	-	3 (7.1)	4 (22.2)	-	-	-
SBP [mmHg]	134 ± 15	138 ± 15	0.414	129 ± 15	133 ± 13	0.421	0.371	0.220
DBP [mmHg]	78 ± 7	80 ± 8	0.436	78 ± 10	81 ± 9	0.175	0.628	0.543
6MWT [m]	328 ± 62	371 ± 63	<0.05	502 ± 66	504 ± 57	0.824	<0.0001	<0.0001
Gait speed [m/s]	0.91 ± 0.17	1.03 ± 0.18	<0.05	1.39 ± 0.18	1.40 ± 0.16	0.824	<0.0001	<0.0001

Abbreviations: BMI body mass index, SBP systolic blood pressure, DBP diastolic blood pressure, 6MWT 6-min walk test. The measurements in groups are compared by the one-way ANOVA or the Mann-Whitney non-parametric test (if the normality assumption is violated).

**Table 2 diagnostics-11-01911-t002:** Medications taken by the patients with peripheral artery disease (%).

Medication	Females *n* = 17	Males *n* = 42	Total *n* = 59
Beta-blockers	7 (41.2%)	17 (40.5%)	24 (40.7%)
ASA	13 (76.5%)	33 (78.6%)	46 (78.0%)
Anticoagulants	7 (41.2%)	16 (38.1%)	23 (39.0%)
Statin	14 (82.4%)	27 (64.3%)	41 (69.5%)
Fenofibrate	0	3 (7.1%)	3 (5.1%)
ACEi	7 (41.2%)	17 (40.5%)	24 (40.7%)
ARB	2 (11.8%)	8 (19.0%)	10 (16.9%)
CCB	6 (35.3%)	14 (33.3%)	20 (33.9%)
Clonidine	2 (11.8%)	2 (4.8%)	4 (6.8%)
Doxazosin	3 (17.6%)	0	3 (5.1%)
Diuretics	7 (41.2%)	15 (35.7%)	22 (37.3%)
Insulin	2 (11.8%)	5 (11.9%)	7 (11.9%)
Metformin	6 (35.3%)	9 (21.4%)	15 (25.4%)
OAD	7 (41.2%)	7 (16.7%)	14 (23.7%)
Bencyclane	2 (11.8%)	8 (19.0%)	10 (16.9%)
Pentoxifylline	4 (23.5%)	8 (19.0%)	12 (20.3%)

Abbreviations: ASA acetylsalicylic acid, ACEi angiotensin converting enzyme inhibitor, ARB angiotensin receptor blocker, CCB calcium channel blockers, OAD oral antidiabetic drugs.

**Table 3 diagnostics-11-01911-t003:** Atherogenic markers (mean ± SD).

Biomarker	Patients		ABI ≤ 0.9 vs. ABI ≤ 0.5	η^2^	Controls	Controls vs. ABI ≤ 0.9	η^2^	Controls vs. ABI ≤ 0.5	η^2^
	ABI ≤ 0.9 *n* = 43	ABI ≤ 0.5 *n* = 16			ABI ≥ 0.9 *n* = 60				
TG [mg/dL]	147 ± 94	129 ± 57	0.878	0.017	140 ± 29	<0.05	0.038	<0.05	0.043
TC [mg/dL]	178 ± 37	185 ± 43	0.838	0.017	224 ± 35	<0.0001	0.287	<0.01	0.128
LDL [mg/dL]	100 ± 33	107 ± 29	0.627	0.013	84 ± 28	<0.05	0.056	<0.05	0.076
HDL [mg/dL]	47 ± 8	45 ± 7	0.393	0.005	72 ± 13	<0.0001	0.605	<0.0001	0.431
non-HDL [mg/dL]	131 ± 38	141 ± 44	0.627	0.013	151 ± 38	<0.01	0.058	0.209	0.008
oxLDL [ng/mL]	467 ± 159	467 ± 137	0.946	0.017	319 ± 284	<0.001	0.131	<0.01	0.080
3NT [nmol/mL]	3.89 ± 2.38	5.33 ± 2.95	0.054	0.048	1.29 ± 0.81	<0.0001	0.532	<0.0001	0.413

Abbreviations: TG triglycerides, TC total cholesterol, LDL low-density lipoproteins, HDL high-density lipoproteins, non-HDL cholesterol calculated by subtracting the HDL value from a TC, oxLDL oxidized low-density lipoprotein, 3NT 3-nitrotyrosine. η^2^ is a measure of effect size. The measurements in groups are compared by the one-way ANOVA or the Kruskal-Wallis non-parametric test non-parametric test (if the normality assumption is violated).

**Table 4 diagnostics-11-01911-t004:** Relationships between neopterin (NPT), C-reactive protein (CRP), ankle-brachial index (ABI), lipoprotein-lipid profile, and progenitor cells in patient with peripheral artery disease (*n* = 59).

	TG [mg/dL]	TC [mg/dL]	LDL [mg/dL]	HDL [mg/dL]	Non-HDL [mg/dL]	EPC [ng/mL]	CD34 [ng/mL]	CD38 [ng/mL]
NPT [nmol/L]	*r_s_* = −0.103 *p* = 0.436	*r_s_* = 0.514 *p* < 0.001	*r_s_* = 0.394 *p* < 0.01	*r_s_* = 0.043 *p* = 0.747	*r_s_* = 0.501 *p* < 0.001	*r_s_* = −0.286 *p* < 0.05	*r_s_* = −0.109 *p* = 0.408	*r_s_* = −0.105 *p* = 0.428
CRP [mg/L]	*r_s_* = −0.132 *p* = 0.318	*r_s_* = 0.394 *p* < 0.01	*r_s_* = 0.363 *p* < 0.01	*r_s_* = 0.023 *p* = 0.861	*r*_s_ = 0.394 *p* < 0.01	*r_s_* = −0.324 *p* < 0.05	*r_s_* = −0.162 *p* = 0.218	*r_s_* = −0.120 *p* = 0.365

*r_s_*—Spearman’s rank correlation coefficient.

**Table 5 diagnostics-11-01911-t005:** Inflammatory markers and progenitor cells (mean ± SD).

Biomarker	Patients		ABI ≤ 0.9 vs. ABI ≤ 0.5	η^2^	Controls	Controls vs. ABI ≤ 0.9	η^2^	Controls vs. ABI ≤ 0.5	η^2^
	ABI ≤ 0.9 *n* = 43	ABI ≤ 0.5 *n* = 16			ABI ≥ 0.9 *n* = 60				
NPT [nmol/L]	8.70 ± 1.95	11.24 ± 3.17	<0.01	0.114	8.15 ± 1.33	0.261	0.003	<0.0001	0.349
CRP [mg/L]	4.37 ± 3.54	10.69 ± 3.52	<0.0001	0.349	2.98 ± 1.94	0.129	0.013	<0.0001	0.464
EPCs [ng/mL]	15.64 ± 6.82	8.90 ± 2.22	<0.0001	0.256	18.90 ± 8.53	<0.05	0.039	<0.0001	0.361
CD34 [ng/mL]	21.30 ± 11.24	14.92 ± 3.69	<0.05	0.078	21.21 ± 10.43	0.730	0.009	<0.05	0.055
CD38 [ng/mL]	0.935 ± 0.582	0.781 ± 0.442	0.344	0.000	1.115 ± 0.793	0.126	0.013	<0.05	0.066

Abbreviations: NPT neopterin, CRP C-reactive protein, EPCs endothelial progenitor cells, CD34 marker of hematopoietic progenitor cells, CD38 marker of non-hematopoietic cells. η^2^ is a measure of effect size. The measurements in groups are compared by the one-way ANOVA or the Kruskal-Wallis non-parametric test non-parametric test (if the normality assumption is violated).

## Data Availability

The data presented in this study are available on request from the corresponding author.

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
