# Peer review of "Assessment of Serum Neopterin as a Biomarker in Peripheral Artery Disease"

_diagnostics, 2021, doi:10.3390/diagnostics11101911_

Round 1
Reviewer 1 Report
Nice review of literature and presentation of your results. for future works I think that a direct comparison with other novel biomarkers will be of great interest.
Author Response
Comments and Suggestions for Authors
Nice review of literature and presentation of your results. for future works I think that a direct comparison with other novel biomarkers will be of great interest.
We greatly appreciate the Reviewer’s comment and the time spent on our manuscript revision. We will bear in mind the remark in our future work.
On balance, the Reviewer’s comments motivated us to re-evaluate our outcomes in order to deliver an improved manuscript.
Reviewer 2 Report
Authors present results of an interesting study on the use of neopterin monitoring in peripheral artery disease. The first papers on the use of neopterin in association with atherosclerosis and myocardial infarction have been published already in 1994 (Melichar B, et al. Increased urinary neopterin in acute myocardial infarction. Clin Chem 1994;40:338-9 - Weiss G, et al. Increased concentrations of neopterin in carotid atherosclerosis. Atherosclerosis 1994;106:263-71). However, even if not generally appreciated, it appears a bit strange to denominate neopterin as a <novel> diagnostic biomarker in this clinical area. At least those first articles need to be mentioned in the paper.
Some of the findings in this study confirm and extend earlier ones described, which exist in the literature since more than a decade by the groups of JC Kaski, e.g., Avanzas P, et al. Elevated serum neopterin predicts future adverse cardiac events in patients with chronic stable angina pectoris. Eur Heart J 2005;26:457-63 – or W Maerz, e.g., Grammer TB, et al. Neopterin as a predictor of total and cardiovascular mortality in individuals undergoing angiography in the Ludwigshafen Risk and Cardiovascular Health study. Clin Chem 2009;55:1135-46. My impression is that those papers need to be acknowledged as well and the new findings need to be put in this context.
However, the new study is important to shed some new light on the use of neopterin in cardiovascular research.
Author Response
Review 2
We greatly appreciate the time spent on our manuscript revision. All of the comments motivated us to re-evaluate our outcomes in order to deliver an improved manuscript.
Comments and Suggestions for Authors
Authors present results of an interesting study on the use of neopterin monitoring in peripheral artery disease. The first papers on the use of neopterin in association with atherosclerosis and myocardial infarction have been published already in 1994 (Melichar B, et al. Increased urinary neopterin in acute myocardial infarction. Clin Chem 1994; 40: 338-9 - Weiss G, et al. Increased concentrations of neopterin in carotid atherosclerosis. Atherosclerosis 1994; 106: 263-71). However, even if not generally appreciated, it appears a bit strange to denominate neopterin as a <novel> diagnostic biomarker in this clinical area. At least those first articles need to be mentioned in the paper.
Thank you for your suggestion. The manuscript title has been rephrased accordingly and now reads as follows: Assessment of serum neopterin as a biomarker in peripheral artery disease
Section 1. Introduction has been enriched with the following information: The first studies on NPT in relation to atherosclerosis and myocardial infarction were already published in the nineties [Melichar et al. Clin Chem 1994, Weiss et al. Atherosclerosis 1994].
Some of the findings in this study confirm and extend earlier ones described, which exist in the literature since more than a decade by the groups of JC Kaski, e.g., Avanzas P, et al. Elevated serum neopterin predicts future adverse cardiac events in patients with chronic stable angina pectoris. Eur Heart J 2005; 26: 457-63 – or W Maerz, e.g., Grammer TB, et al. Neopterin as a predictor of total and cardiovascular mortality in individuals undergoing angiography in the Ludwigshafen Risk and Cardiovascular Health study. Clin Chem 2009; 55: 1135-46. My impression is that those papers need to be acknowledged as well and the new findings need to be put in this context.
Following the Reviewer’s suggestion, Section 1. Introduction has been enriched with the following information: According Avanzas et al. [Eur Heart J 2005] elevated serum NTP predicts future adverse cardiac events in patients with chronic stable angina pectoris. Furthermore, some studies demonstrated an association of NPT with the ankle-brachial index (ABI) and increased mortality of patients with vascular calcification or patients undergoing angiography [10,15, Grammer et al. Clin Chem 2009].
This manuscript is a resubmission of an earlier submission. The following is a list of the peer review reports and author responses from that submission.